# Gp41-targeted antibodies restore infectivity of a fusion-deficient HIV-1 envelope glycoprotein

**Vinita R. Joshi**[1,2], **Ruchi M. Newman**[1¤a], **Melissa L. Pack**[1¤b], **Karen A. Power**[1], **James B. Munro**[3], **Ken Okawa**[1], **Navid Madani**[4], **Joseph G. Sodroski**[4,5,6], **Aaron G. Schmidt**[1,6], **Todd M. Allen**[1] *

**1** Ragon Institute of MGH, MIT and Harvard, Cambridge, Massachusetts, United States of America,
**2** Department of Virology, Harvard Medical School, Boston, Massachusetts, United States of America,
**3** Department of Microbiology and Physiological Systems, University of Massachusetts Medical School, Worcester, Massachusetts, United States of America, **4** Department of Cancer Immunology and Virology, Dana-Farber Cancer Institute, Boston, Massachusetts, United States of America, **5** Department of Immunology and Infectious Diseases, Harvard T.H. Chan School of Public Health, Boston, Massachusetts, United States of America, **6** Department of Microbiology, Harvard Medical School, Boston, Massachusetts, United States of America

¤a  Current address: Unum Therapeutics, Cambridge, Massachusetts, United States of America.
¤b  Current address: Department of Computation Biology, Foundation Medicine, Inc., Boston, Massachusetts, United States of America.
* tallen2@mgh.harvard.edu

**Data Availability Statement:** All env sequence files are available from the GenBank database (accession numbers MT023027, MT023028,

## Abstract

The HIV-1 envelope glycoprotein (Env) mediates viral entry via conformational changes associated with binding the cell surface receptor (CD4) and coreceptor (CCR5/CXCR4), resulting in subsequent fusion of the viral and cellular membranes. While the gp120 Env surface subunit has been extensively studied for its role in viral entry and evasion of the host immune response, the gp41 transmembrane glycoprotein and its role in natural infection are less well characterized. Here, we identified a primary HIV-1 Env variant that consistently supports >300% increased viral infectivity in the presence of autologous or heterologous HIV-positive plasma. However, in the absence of HIV-positive plasma, viruses with this Env exhibited reduced infectivity that was not due to decreased CD4 binding. Using Env chimeras and sequence analysis, we mapped this phenotype to a change Q563R, in the gp41 heptad repeat 1 (HR1) region. We demonstrate that Q563R reduces viral infection by disrupting formation of the gp41 six-helix bundle required for virus-cell membrane fusion. Intriguingly, antibodies that bind cluster I epitopes on gp41 overcome this inhibitory effect, restoring infectivity to wild-type levels. We further demonstrate that the Q563R change increases HIV-1 sensitivity to broadly neutralizing antibodies (bNAbs) targeting the gp41 membrane-proximal external region (MPER). In summary, we identify an HIV-1 Env variant with impaired infectivity whose Env functionality is restored through the binding of host antibodies. These data contribute to our understanding of gp41 residues involved in membrane fusion and identify a mechanism by which host factors can alleviate a viral defect.

MT023029, MT023030, MT023031, MT023032, MT023033).

**Funding:** This study was supported by funding through the Ragon Institute of MGH, MIT and Harvard (http://www.ragoninstitute.org/) and the MGH Scholars program (https://ecor.mgh.harvard.edu/Default.aspx?node_id=338) (TMA), and NIH grant P01 AI04715 (TMA). The funders had no role in study design, data collection and analysis, decision to publish, or preparation of the manuscript.

**Competing interests:** The authors have declared that no competing interests exist.

## Author summary

HIV-1 Env consists of the surface subunit (gp120) and the transmembrane subunit (gp41). Receptor and coreceptor binding of the Env trimer triggers structural rearrangements within gp41 leading to the formation of a six-helix bundle. These gp41 conformational changes are critical to membrane fusion and viral entry. We have identified a change in gp41, Q563R, that disrupts this six-helix bundle formation, negatively affecting viral entry and infection. Surprisingly, the humoral immune response to HIV-1 counter-intuitively overcomes the conformational disruption, rescuing a viral defect to allow for increased viral infection. We also demonstrate that the Q563R change confers increased sensitivity to broadly neutralizing antibodies targeting the gp41 MPER. Thus, the Q563R change impedes six-helix bundle formation and virus entry, but these Env functions can be restored by naturally occurring antibodies. Our data further suggests that R563 might enhance the binding of MPER-specific bNAbs, and therefore could be useful in an Env immunogen.

## Introduction

HIV-1 Env consists of two subunits: the surface-exposed glycoprotein gp120, which contains the receptor (CD4) and coreceptor (CCR5 or CXCR4) binding sites, and the transmembrane subunit gp41, which is critical for virus-cell membrane fusion [1–5]. The ectodomain of gp41 contains two helical heptad repeats at the N- and C-termini (HR1 and HR2, respectively) connected by a disulfide loop [6, 7]. Sequential binding of HIV-1 to its receptor and coreceptor are the first steps critical for viral entry [1, 8–11], whereby CD4 binding triggers a series of conformational changes in Env involving both gp120 and gp41. In gp120, CD4 binding induces rearrangement of the variable loops V1/V2 and formation of the bridging sheet, which allows for repositioning of the V3 loop to facilitate coreceptor binding [1, 8]. In gp41, CD4 binding triggers exposure of the buried N-terminal HR1 domain and its formation into a trimeric coiled coil structure [12]. Upon receptor and coreceptor binding, the fusion peptide in gp41 is inserted into the cell membrane and the HR1 and HR2 regions rearrange to form the six-helix bundle, which is critical for creation of the fusion pore to enable membrane fusion and efficient viral entry [13–15].

Alteration of multiple Env amino acid residues can affect viral infection by modulating the efficiency of the conformational changes required for entry. In the case of gp41, these can affect gp120-targeted CD4 binding, as well as gp41-mediated fusion [16–18]. For instance, residues A582, Q577 and E560 in HR1 decrease conformational reactivity upon CD4 binding, while HR1 residue L587 destabilizes a CD4-bound conformation [16, 17]. Since gp41 is the necessary component for virus-cell membrane fusion, it is not surprising that changes in gp41 residues can also modulate membrane fusion and consequently, viral infection. The gp41 six-helix bundle, which is thought to be the core mediator of the membrane fusion process, contains a central coiled coil of HR1 trimers, against which the HR2 regions from each gp41 protomer interact in an antiparallel manner [13]. The stability of the six-helix bundle is dependent in part upon multiple interactions between HR1 and HR2 residues. Specific HR1 residues that interact with HR2 have been identified and the effects of changing these residues to alanine have been tested in the context of virus-cell fusion and infection with varying outcomes [17, 19]. Notably, the amino acid residue glutamine (Q) at position 563 (HXB2 numbering) in HR1 interacts with isoleucine (I) at positions 642 and 646 of HR2. Q563 also interacts with HR1 intra-domain residues, potentially forming a hydrogen bond with Q577, or by interacting with

L565 and affecting Q562's interaction with I559 via its polar side chain [16, 19–21]. Previous reports also show that changing Q563 to alanine (Q563A) modestly reduces viral fusion, whereas alteration to arginine (Q563R) causes a dramatic loss of infectivity [17, 20, 22].

Env mutagenesis during HIV-1 evolution and the generation of escape variants in response to immune pressure can be deleterious for the virus [23]. While Env changes generally result in a fitness cost, the pliability of Env often enables it to mitigate these effects through various compensatory mutations [24–26]. However, in some cases, HIV-1 has demonstrated the ability to utilize host factors to overcome these defects. A study examining the development of resistance in an individual treated with the T20 fusion inhibitor led to the identification of a viral variant that harbored T20-escape changes within gp41. Interestingly, this dominant variant was not only resistant to T20, but also critically dependent on the inhibitor for its replication [27]. Here, three escape-related changes in gp41 rendered Env in a hyperfusogenic conformation, which prematurely triggered gp41 and led to a loss in infectivity. However, the peptide T20 was proposed to act as a "safety pin" to prevent premature activation of this escape variant, thus restoring viral infectivity. More broadly, this study demonstrated that other fusion inhibitors designed to impede the formation of the six-helix bundle also provided an advantage to the hyperfusogenic Env protein, presumably through the same mechanism [27, 28]. These studies highlight a phenomenon whereby specific HIV-1 escape variants result in a fitness cost, but the viral defect is surprisingly alleviated in the presence of specific antiviral factors.

In the course of characterizing the longitudinal evolution of Env in a subject exhibiting a broadly neutralizing antibody response to HIV-1, we have identified a naturally occurring Env variant containing a Q563R change in gp41. Here we demonstrate that the Q563R change reduces HIV-1 infectivity by disrupting viral membrane fusion, and that Envs harboring this alteration are dependent on gp41-targeted antibodies to overcome this viral defect. Additionally, we evaluate the effect of this single Q563R substitution on Env conformation and recognition by MPER-targeted bNAbs.

## Results

### A primary HIV-1 Env (E1) glycoprotein supports increased viral infection in the presence of HIV-positive plasma

Subject 653116 (initially described as AC053) was previously identified to develop neutralizing antibody breadth against a panel of HIV-1 isolates [29]. ln the course of studying late-stage (6.5 years post-infection (ypi)) Env clones on pseudoviruses, we identified a clone (termed E1 for Enhancing clone #1) that exhibited a >300% increase in infection in the presence of autologous plasma (**Fig 1A**). This increased infection also occurred in the presence of heterologous plasma (Heterologous HIV+ Plasma 1) from an unrelated, HIV-1-infected individual (**Fig 1B**). In examining 9 other viral isolates, including 6 primary Envs from individual 653116, this HIV-1-positive plasma-dependent increase in infection was found to be unique to Env E1 (**Fig 1C**). Moreover, autologous plasma from both 4.6 ypi (**Fig 1A**) and 2 ypi (**Fig 1C**) supported this effect. In contrast, in the presence of two HIV-negative plasma samples, Env E1 supported levels of infection comparable to those of non-enhancing (NE) Env clones NE1 (6.5 ypi) and NE2 (0.8 ypi) from subject 653116 (**Fig 1D and Fig 1E**). These data indicate the unique ability of Env clone E1 to mediate enhanced viral infectivity that was dependent upon HIV-1-positive plasma.

We investigated the fraction of plasma facilitating this increased infection by viruses with Env E1. We found that purified IgG from subject 653116's autologous plasma was sufficient to recapitulate this increased infection (**Fig 1F**). Next, we fractionated this plasma using magnetic beads coated with the HIV-1 Env RSC3core protein which uniquely binds gp120-specific

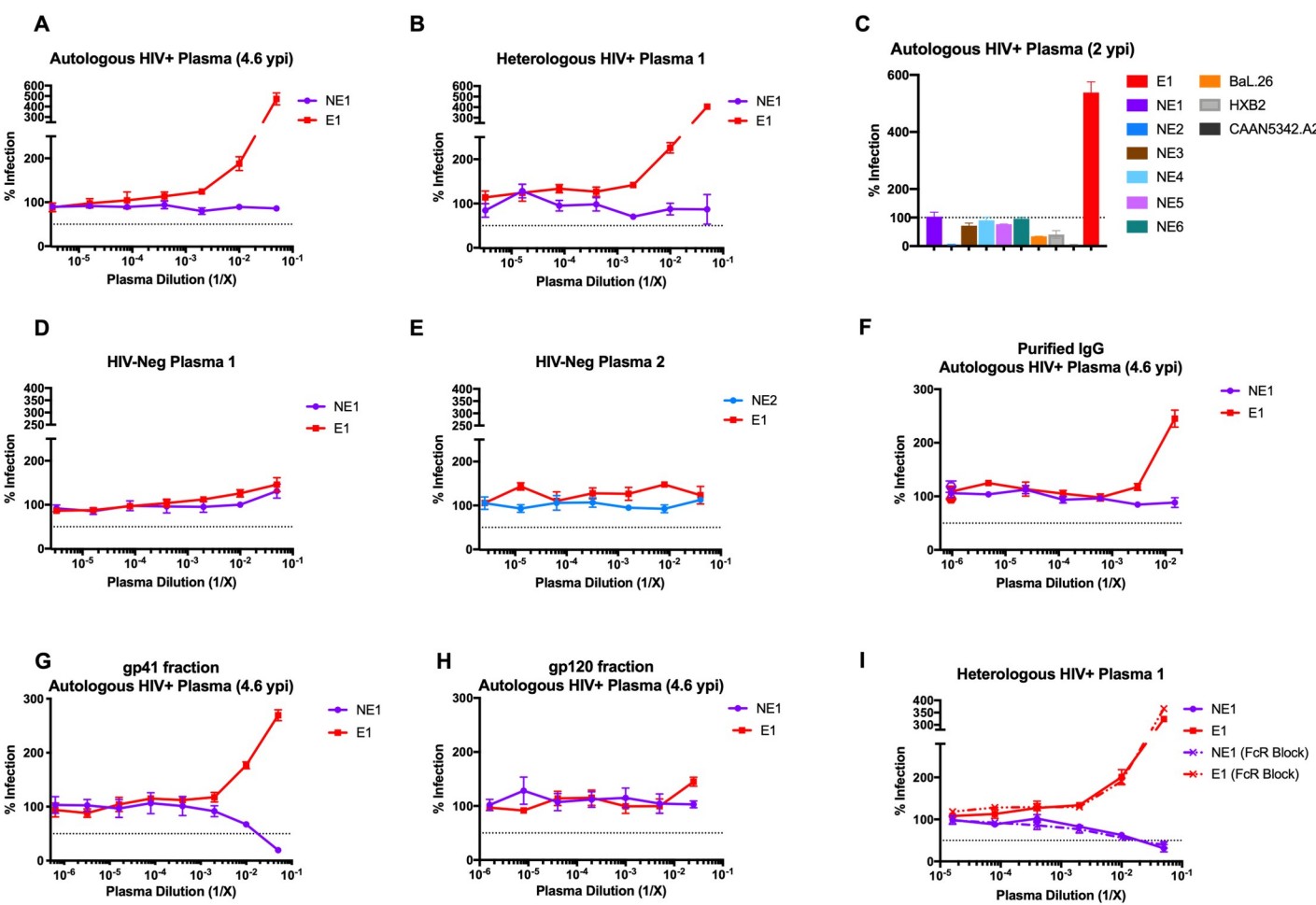

**Fig 1. Identification of a primary HIV-1 Env clone (E1) that supports enhancement of infection in the presence of HIV-1-positive plasma.** Infection of TZM-bl cells by two viruses with Env clones from individual 653116 in the presence HIV-1-positive **(A)** autologous plasma and **(B)** heterologous plasma is shown. Env glycoprotein exhibiting increased virus infectivity (E1) was compared to a contemporary Env glycoprotein from the same time point that does not exhibit increased virus infectivity (NE1). **(C)** Five additional autologous Env clones from individual 653116 that do not exhibit enhanced virus infectivity are designated NE2-NE6. **(D, E)** Infection of pseudoviruses with E1, NE1 or NE2 Env was tested in the presence of two HIV-1-negative plasma samples. **(F-H)** Infection by E1 and NE1 was tested in the presence of **(F)** purified IgG, **(G)** gp120-binding depleted antibodies, and **(H)** gp120-binding antibodies from HIV-1-positive plasma. **(I)** Infection mediated by Envs E1 and NE1, with and without Fc-receptor blocking, in the presence of heterologous HIV-1-positive plasma 1. All assays were conducted in triplicate. The % infection reported was observed after incubation of TZM-bl cells with a similar number of infectious units ($10^5$ RLU) of viruses. Data are represented as mean values; error bars represent SEM. The dotted line indicates 50% infection, except in the case of **(C)**, where it indicates 100% infection.

antibodies [30]. The flow-through plasma fraction, confirmed by ELISA to contain gp41-binding antibodies **(S1 Fig),** increased infection of viruses with Env E1 **(Fig 1G)**. In contrast, gp120-bound antibodies eluted from the RSC3core protein did not support increased infection by E1 viruses **(Fig 1H)**. These observations suggest that gp41-targeted antibodies mediate the increase in infection by viruses with Env E1. Finally, enhancement of *in vitro* infection has thus far been attributed to either complement-mediated enhancement [31] or Fc-receptor (FcR)-mediated enhancement [32, 33]. The use of heat-inactivated plasma in our assays, which destroys complement [34], eliminates the potential for complement-mediated enhancement accounting for the increased infectivity. Similarly, Fc-receptor blocking using the Fc receptor blocking solution (Human TruStain FcX) did not alter the increased infection seen for viruses with Env E1 in the presence of HIV-positive plasma **(Fig 1I)**. Taken together, our data indicate

that the increase in infection exhibited by viruses with Env E1 is dependent on the presence of gp41-targeted antibodies, and is independent of complement or Fc receptors.

## Reduced infectivity of viruses with Env E1 is not due to differential binding of CD4 or CD4-induced changes

As seen above, the infectivity of the Env E1 virus in the presence of HIV-positive plasma was higher than Env E1 virus infectivity in the absence of plasma (**Fig 1**). For these assays, pseudovirus input was normalized to dilutions resulting in 100,000 relative luciferase units (RLU) in TZM-bl cells, which was always titered in the absence of any plasma or inhibitors. This method of titration does not account for the presence of non-infectious Env. Thus, normalizing input to RLU may result in unequal Env particle input, based on differences in the infectivity of each Env.

To determine whether the increased infection of viruses with the Env E1 in the presence of HIV-positive plasma might be due to intrinsic differences in infectivity, we normalized pseudovirus input by reverse transcriptase (RT) activity and quantitated infection of TZM-bl cells. Interestingly, Env E1 viruses were found to be >3-fold less infectious than Env NE1 viruses (**Fig 2A**). To test if this reduced infectivity of Env E1 viruses was due to impaired binding to the CD4 receptor, we compared the neutralization sensitivity of E1 and NE1 viruses in the presence of both CD4-Ig (**Fig 2B**) as well as soluble CD4 (sCD4) (**Fig 2C**). No differences in neutralization by CD4-Ig or sCD4 were observed between E1 and NE1 viruses, indicating that the decrease in infectivity of viruses with Env E1 was not due to reduced binding to CD4. We further confirmed this result using the CD4-mimetic compound DMJ-II-121, which binds the

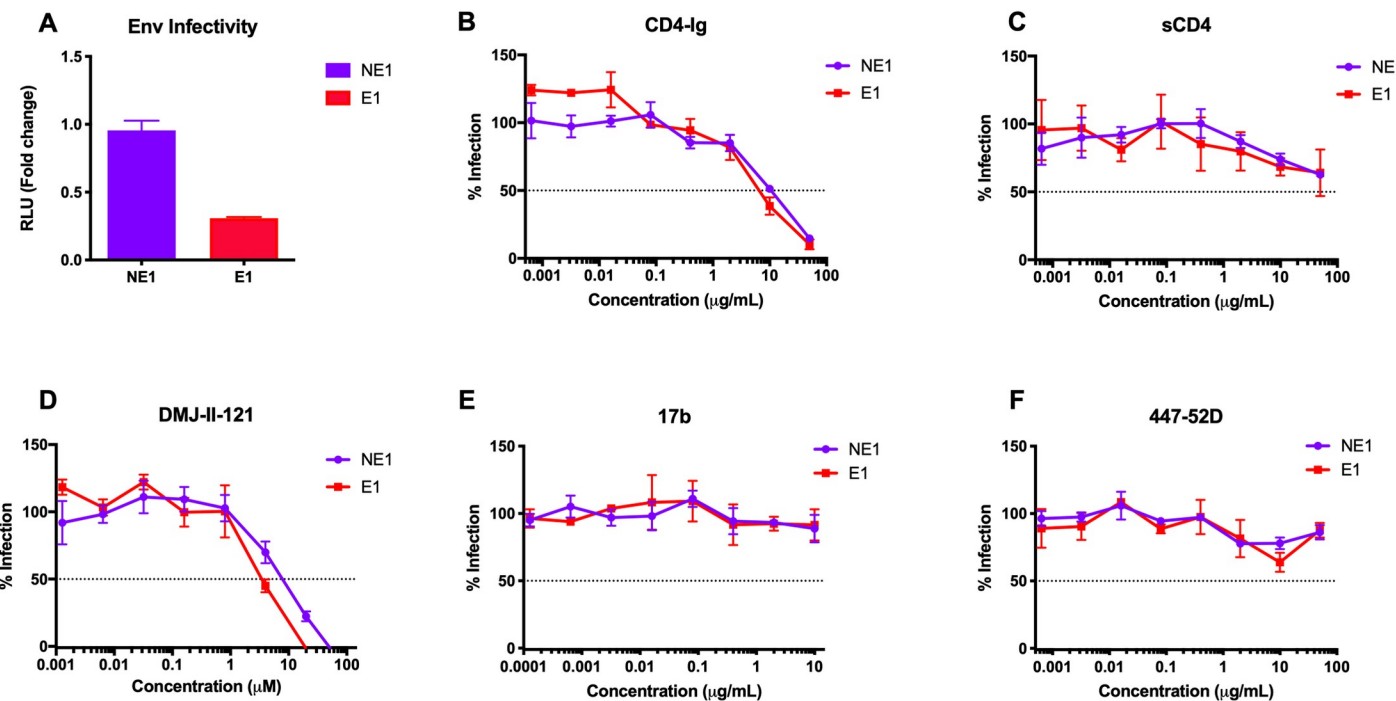

**Fig 2. Reduced infectivity of viruses with Env E1 is not due to differential binding of CD4 or CD4-induced changes.** Infection of TZM-bl cells by NE1 and E1 viruses (**A**) by normalizing pseudovirus input to RT activity. In **B-F**, 100,000 RLU of the E1 and NE1 pseudoviruses was incubated with TZM-bl cells in the presence of (**B**) CD4-Ig, (**C**) soluble CD4 (sCD4), (**D**) CD4-mimetic DMJ-II-121, and the CD4 induced antibodies (**E**) 17b and (**F**) 447-52D. All assays were conducted in triplicate. Data are represented as mean values; error bars indicate SEM. The dotted line indicates 50% infection.

CD4-binding site of gp120 and triggers the CD4-induced (CD4i) conformation (**Fig 2D**). Next, to determine whether the reduced infectivity of Env E1 was associated with conformational changes induced by CD4 binding, we tested neutralization of E1 and NE1 viruses by the CD4i antibodies 17b and 447-52D. Both E1 and NE1 viruses were largely resistant to neutralization by these CD4i antibodies, as expected for primary HIV-1 (**Fig 2E and 2F**). Thus, differential exposure of CD4i epitopes is not a factor contributing to Env E1's lower infectivity. Taken together, these data indicate that viruses with Env E1 are less infectious than viruses with the contemporary Env NE1, but that this reduced infectivity is not due to differential binding of CD4 or the premature induction of CD4i conformational changes.

### A single-residue change Q563R in gp41 of Env E1 is responsible for the increased infectivity in the presence of HIV-positive plasma

To determine the Env region responsible for the increased infectivity of E1 viruses in the presence of HIV-1-positive plasma, we generated chimeric Envs between E1 and NE1 and compared the infectivity of the resulting viruses in the presence of HIV-1-positive plasma. While Env chimeras containing Env E1 V1/V2 loops as well as the C2, V3, C3, V4, C4 and C5 regions failed to exhibit the increased infection phenotype (**S2 Fig**), the Env E1 gp41 region when engrafted into the Env NE1 fully recapitulated this phenotype (**Fig 3A**). Sequence analysis of

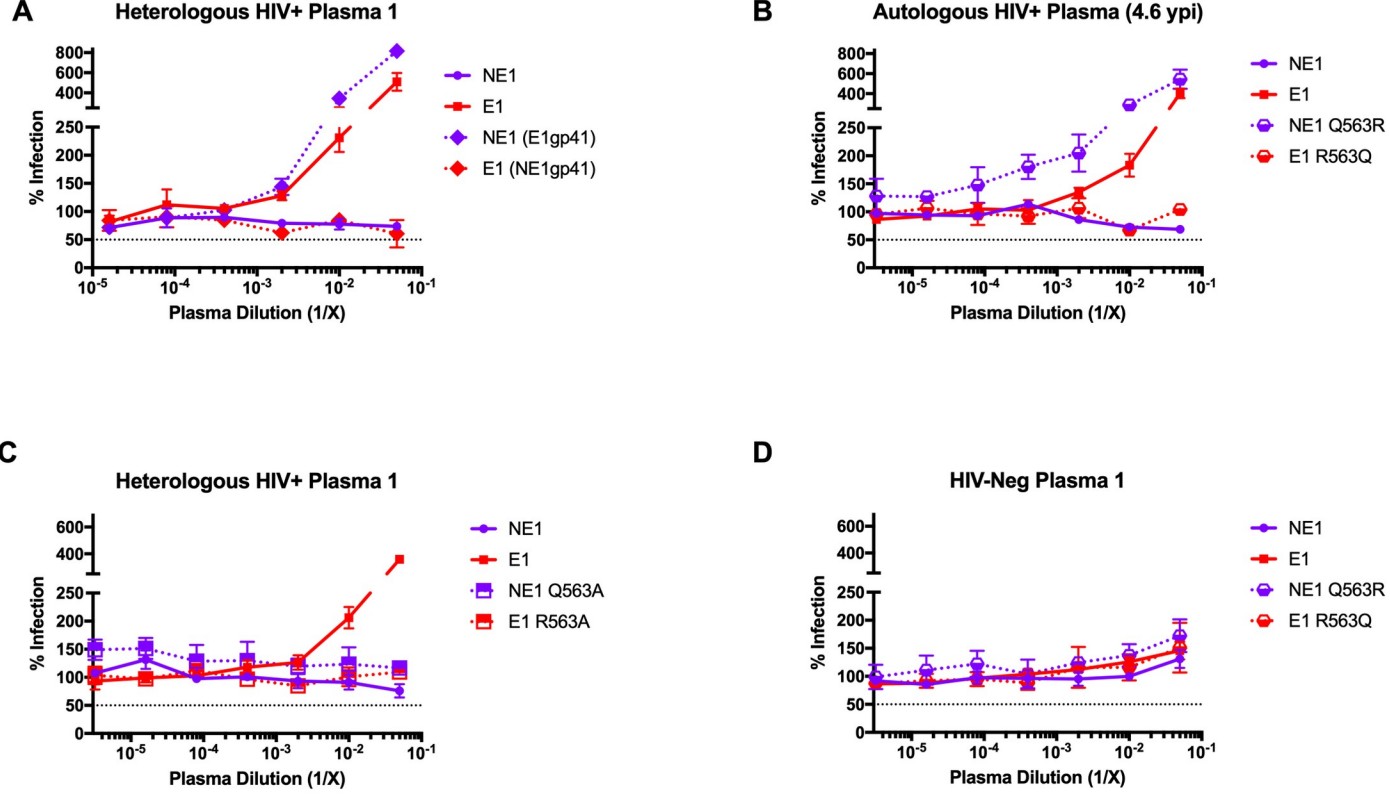

**Fig 3. A single-residue change in the gp41 region (Q563R) of Env E1 is responsible for the increased infection observed specifically with HIV-positive plasma.** Comparative infection of TZM-bl cells in the presence of HIV-1-positive plasma by (**A**) E1 and NE1 viruses, as well as Env E1/NE1 gp41 chimeras, (**B**) E1 and NE1 viruses with or without changes at residue 563 to Q (Glutamine) or R (Arginine), respectively; and by (**C**) E1 and NE1 viruses with or without a 563A (Alanine) substitution. (**D**) Comparative infection of TZM-bl cells in the presence of HIV-1-negative plasma by E1 and NE1 viruses with or without changes at residue 563 to Q (Glutamine) or R (Arginine), respectively. All assays were done in triplicate. Data are represented as mean values; error bars indicate SEM. The dotted line indicates 50% infection.

the gp41 regions in Env E1 compared to each of the non-enhancing Envs (NE1, NE2, NE3, NE4, NE5 and NE6) identified a glutamine to arginine (Q→R) change at position 563 that was unique to Env E1 (**S4 Fig**). The glutamine at position 563 in the HR1 region is highly conserved (~1.5% variability) (www.hiv.lanl.gov) [35], and previous structure studies of gp41 propose a role for this residue in viral entry via its interaction with residues in the HR2 region of gp41 [16, 20].

Next, we altered Q563 in Env NE1 to arginine (NE1 Q563R), and similarly altered R563 to glutamine in Env E1 (E1 R563Q). Viruses with the Q563R variant of Env NE1 exhibited the same increased infection as Env E1 in the presence of autologous HIV-1-positive plasma. In contrast, viruses with the Env E1 R563Q Env did not exhibit the plasma-dependent increase in infection (**Fig 3B**). We further tested the infectivity of these viruses in the presence of heterologous plasma from another unrelated HIV-1 infected individual (Heterologous HIV+ Plasma 2) (**S3 Fig**). Unlike Heterologous HIV+ Plasma 1 (**Fig 3A**), Heterologous HIV+ Plasma 2 exhibited neutralization of the Env NE1 virus. Both Env E1 and Env NE1 Q563R viruses were not neutralized by this plasma, but instead exhibited increased infectivity. On the same lines, the R→Q substitution in Env E1 R563Q virus did not support increased infectivity, and reverted to the phenotype seen with Env NE1 virus, indicating that the Q563R change mediated increased infectivity even in the presence of neutralizing antibodies in HIV positive plasma (**S3 Fig**). To test whether arginine at position 563 was specifically critical for this phenotype, we changed this residue in both Env E1 and NE1 to alanine (E1 Q563A and NE1 Q563A, respectively). Viruses with Env NE1 Q563A failed to exhibit the increase in infection seen with Q563R. Similarly, the Env E1 R563A change also eliminated this effect, indicating that the presence of arginine at position 563 is important for this phenotype (**Fig 3C**). As a control, we verified that neither the arginine-expressing Env NE1 (NE1 Q563R) nor the glutamine-expressing Env E1 (E1 R563Q) mutants exhibited increased infectivity in the presence of HIV-negative plasma (**Fig 3D**). Therefore, through the testing of chimeric Envs and point mutants we have identified a single-residue change in gp41, Q563R, that is responsible for the HIV-1-positive plasma-dependent increase in infection observed for viruses with Env E1.

## The Q563R change dramatically decreases viral infectivity by disrupting membrane fusion and six-helix bundle formation

To test whether the increased infection of NE1 Q563R viruses in the presence of HIV-1-positive plasma was a result of the same mechanism affecting E1 virus infectivity, we normalized pseudovirus input to RT activity and tested infection of TZM-bl cells. As expected, the NE1 Q563R virus was less infectious than the NE1 virus (>20-fold lower), while mutant Env E1 R563Q demonstrated a >5-fold increase in infectivity compared to viruses with wild-type Env E1 (**Fig 4A**). Based on these data, and the location of residue 563 in HR1, we hypothesized that the decreased infectivity of E1 and NE1 Q563R viruses is due to pre-fusion disruption of the six-helix bundle formation. To explore this hypothesis, we tested virus-cell fusion using the β-lactamase–Vpr (Blam-Vpr) assay [36]. While Env E1 showed a slight decrease in viral fusion compared to NE1, NE1 Q563R showed a dramatic loss in viral infectivity, corroborating the significant loss in Env infectivity seen with this Env. Similarly, membrane fusion of Env E1 R563Q was significantly improved over that of Env E1, consistent with the ability of this R→Q change to alleviate the defect in viral fusion and restore infectivity (**Fig 4B**).

We further tested inhibition of these Envs using the C34 fusion inhibitor peptide, which targets the HR1 region in Env and inhibits the formation of the six-helix bundle [37]. Here, resistance to C34 might indicate a kinetically less accessible or more stable six-helix bundle, while increased sensitivity to C34 might indicate greater accessibility to the HR1 region, for

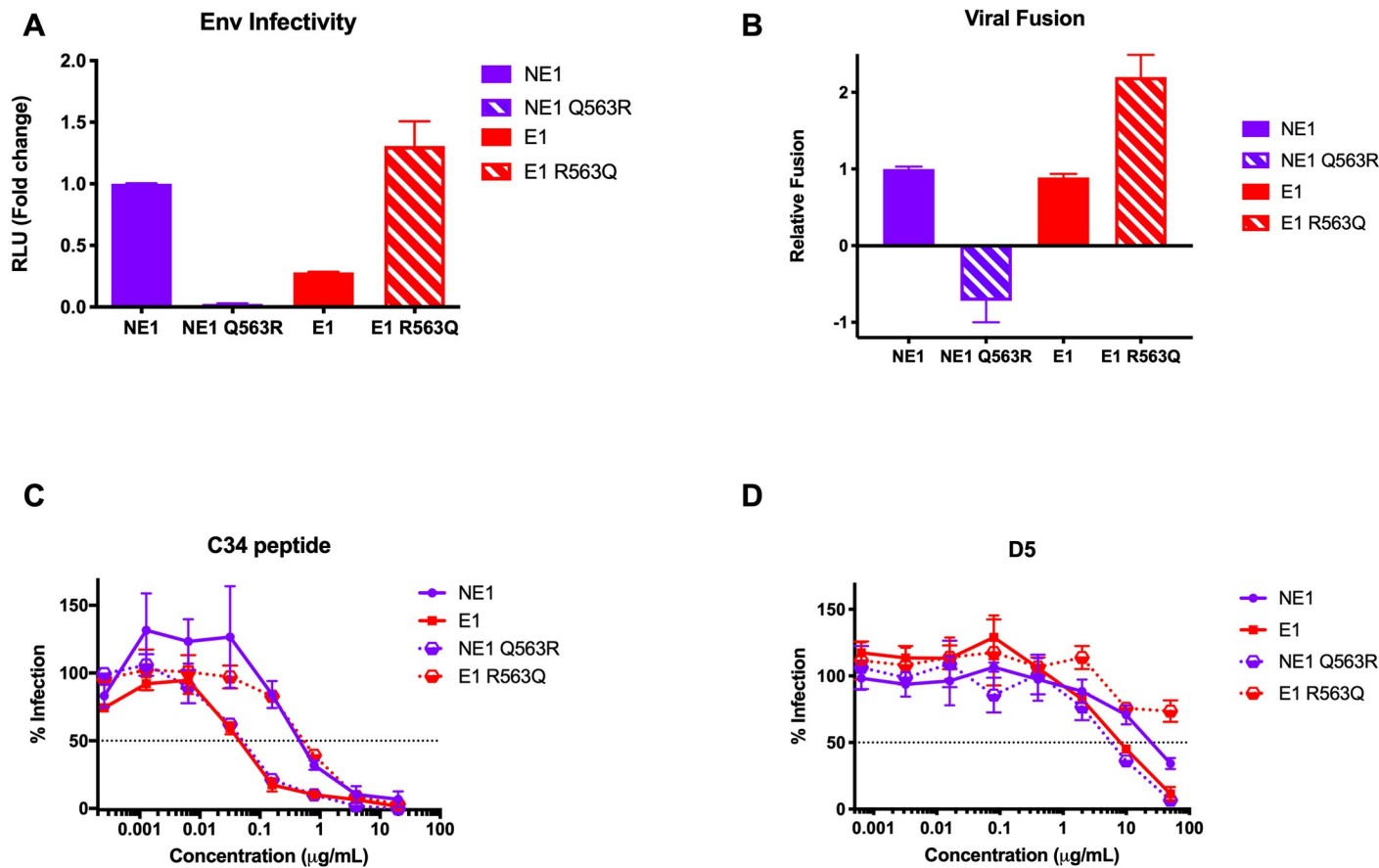

**Fig 4. Q563R reduces Env infectivity by disrupting membrane fusion and six-helix bundle formation.** (A) Relative infectability of TZM-bl cells by viruses with NE1, E1, NE1 Q563R and E1 R563Q Envs, normalized for RT activity. (B) Relative fusion of TZM-bl cells with NE1, E1, NE1 Q563R and E1 R563Q Blam-Vpr viruses, normalized to Env NE1 virus levels. Inhibition of infection of TZM-bl cells by viruses with these Envs in the presence of the (C) C34 peptide and (D) D5 monoclonal antibody. All assays were done in triplicate. Data are represented as mean values; error bars indicate SEM. The dotted line indicates 50% infection.

example due to a deformed or unstable six-helix bundle. We found that NE1 and E1 R563Q viruses were inhibited by C34 at an $IC_{50}$ of 0.4μg/mL and 0.6μg/mL, respectively. In contrast, the Q563R change rendered both E1 and NE1 Q563R viruses ~10-fold more sensitive to C34 inhibition, with $IC_{50}$ values of 0.05μg/mL and 0.04μg/mL, respectively (**Fig 4C**). Thus, the Q563R substitution in the E1 and NE1 Q563R Envs increased accessibility of C34 to gp41, potentially indicating disruption of the six-helix bundle.

To confirm this hypothesis, we tested inhibition of these Envs by the monoclonal antibody D5, which is known to preferentially bind the pre-fusion three-helix trimer conformation consisting of a coiled-coil of the three HR1 regions [38–40]. As the six-helix bundle is formed through interaction of HR2 residues with this HR1 three-helix structure, the epitope for D5 normally becomes occluded during six-helix bundle formation, resulting in reduced binding. Thus, similar to C34 inhibition, increased binding of D5 to Env might indicate a less stable six-helix bundle, increasing the accessibility of the three-helix bundle intermediate. As seen in **Fig 4D**, both NE1 and E1 R563Q viruses were relatively resistant to D5 inhibition ($IC_{50}$ values of 26μg/mL and >50μg/mL respectively). In contrast, the presence of the Q563R change increased virus inhibition by D5 ($IC_{50}$ values of 8μg/mL for Env E1 and 0.6μg/mL for Env NE1 Q563R). These data suggest that the changes in E1 and NE1 Q563R virus infectivity observed

are modulated by disrupting normal six-helix bundle formation and subsequent fusion, thereby affecting viral entry. As such, these data suggest that the gp41-targeted fraction in plasma that rescues this deficient phenotype for Env E1 infection (**Fig 1G**) may be due to gp41-specific antibodies stabilizing the process of formation of a stable six-helix bundle.

## Anti-cluster I antibodies restore the Q563R-mediated infectivity defect

To determine the nature of gp41-binding antibodies that are responsible for stabilizing the six-helix bundle, we tested a panel of previously described monoclonal antibodies that target HIV-1 gp41. Antibodies 240-D, 246-D and F240 bind overlapping cluster I epitopes in the C-terminal region of HR1 and the C-C loop between HR1 and HR2 [41, 42]; antibody T32 binds a distinct epitope containing the C-C loop towards HR2; and antibodies NC-1 and 98–6 bind HR2 [41, 43]. We tested the anti-cluster I antibodies for their effect on infectivity by viruses with Env NE1, Env E1 and their respective single-residue 563 arginine/glutamine mutants. HR1-targeting monoclonal antibodies 240-D, 246-D and F240 restored infectivity of E1 and NE1 Q563R viruses to levels observed with HIV-1-positive plasma (**Fig 5A, 5B and 5C**). T32, which has previously been described as an HR1 targeting antibody, did not mediate this effect (**Fig 5D**). To confirm the specificity of gp41-binding antibodies that mediate increased infectivity of viruses with Q563R-containing Envs, we tested infection in the presence of HR2-binding antibodies 98–6 and NC-1. Neither of these HR2-binding antibodies resulted in higher infection by E1 or NE1 Q563R viruses (**S5A Fig** and **S5B Fig.**). As a control, antibody 902090 targeting the V2-loop showed no effect (**S5C Fig**). The epitope-specificity of these anti-cluster I antibodies narrowed down the critical region for these restorative antibodies to the $_{592}$LLGIW$_{596}$ epitope within the gp41 C-C loop (**S5D Fig**).

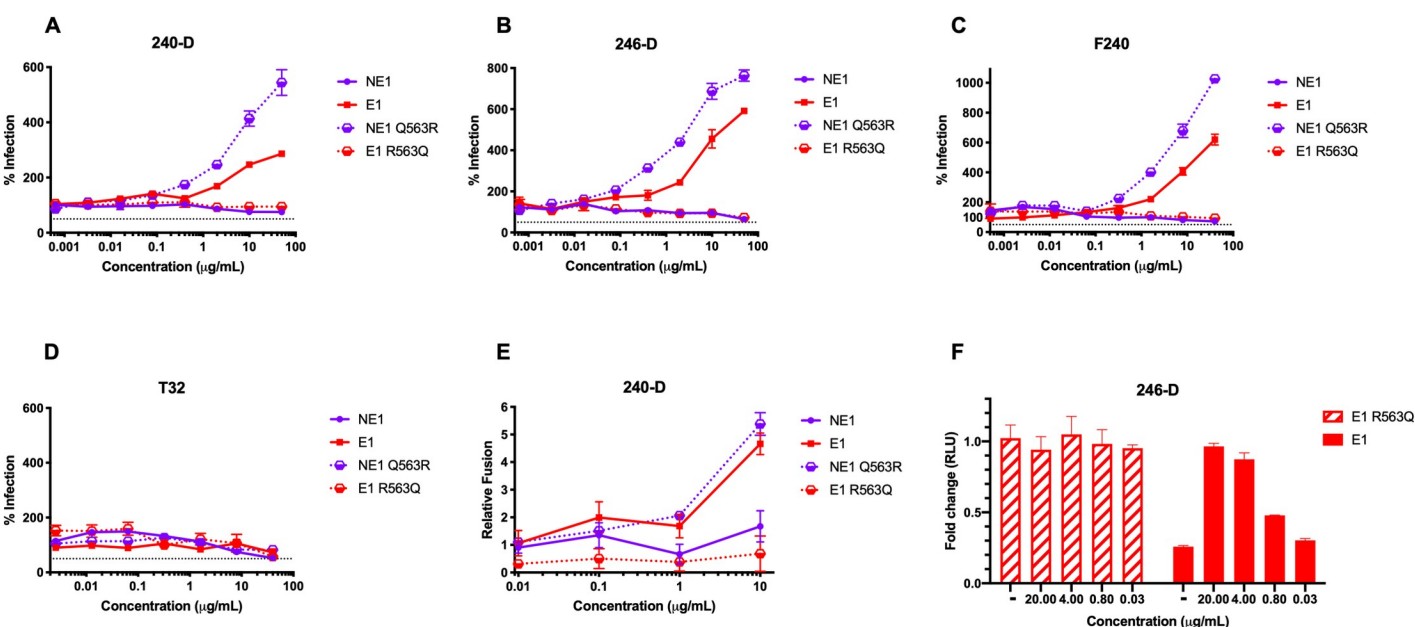

**Fig 5. Monoclonal antibodies targeting HR1 and the C-C loop of gp41 restore infectivity defects observed with Q563R.** Infection of TZM-bl cells by NE1, NE1 Q563R, E1 and E1 R563Q viruses was tested in the presence of anti-cluster I antibodies (**A**) 240-D, (**B**) 246-D, (**C**) F240 and (**D**) T32. (**E**) Relative fusion of Blam-Vpr viruses with NE1, NE1 Q563R, E1 and E1 R563Q Envs with TZM-bl cells in the presence of HR-1-targeting mAb 240-D is shown. (**F**) Fold change in infection of TZM-bl cells by E1 and E1 R563Q viruses with different amounts of 246-D. Values are normalized to levels seen for E1 R563Q virus in the absence of antibody; data are represented as fold-change compared to E1 R563Q virus infectivity. All assays were conducted in triplicate. Data are represented as mean values; error bars indicate SEM. The dotted line indicates 50% infection.

To explore whether the HR1-targeting antibodies could mediate increased infectivity by rescuing virus-cell membrane fusion, we measured fusion of Blam-Vpr viruses expressing NE1, NE1 Q563R, E1 or E1 R563Q Envs in TZM-bl cells in the presence of mAb 240-D. Virus-cell fusion was quantified and plotted as fold change relative to fusion in the absence of antibody. As observed for viral infectivity, at a concentration of 10μg/mL, mAb 240-D exhibited a 5-fold increase in virus-cell fusion of Q563R-expressing Env viruses. Similar to the increased viral infectivity, this effect was dose-dependent (**Fig 5E**). These results indicate that the antibody-dependent increased infectivity may be due to restored virus-cell membrane fusion.

Since gp41-targeting antibodies themselves are known to enhance overall infection, we further sought to confirm that the mAb-mediated increase in infection indeed constituted a rescue of an infectivity defect versus an enhancement phenotype above wild-type levels [44]. We tested infectivity of Env E1 and E1 R563Q pseudovirus, normalized by RT activity, in the presence of different concentrations of mAb 246-D. As shown earlier, in the absence of antibody, the E1 virus was 5-fold less infectious than the E1 R563Q virus. E1 R563Q virus infectivity was unaffected by the addition of mAb 246-D at different concentrations. In contrast, at high concentrations of mAb 246-D (20μg/mL), infectivity levels of the E1 virus were restored to those seen with the E1 R563Q virus. A 5-fold dilution (4μg/mL) of mAb 246-D exhibited similar levels of infectivity of the E1 virus, indicating plateauing of infection by saturating levels of antibody. The increase in E1 virus infectivity up to wild-type levels (and not higher) indicates that this antibody rescues an infectivity defect, versus generally enhancing infection (**Fig 5F**). Thus, our results suggest that specific antibodies targeting a narrow $_{592}$LLGIW$_{596}$ epitope within the gp41 C-C loop may help stabilize six-helix bundle formation and restore virus infectivity disrupted by the Q563R change (**S5D Fig**).

## Q563R allows for better binding of MPER bNAbs

The Env trimer has been extensively studied as a potential immunogen to induce broadly neutralizing antibodies [45, 46]. Env trimer instability and heterogeneity makes the trimer an inconsistent target to induce such bNAb responses. The use of BG505 SOSIP.664 highlights the importance of stabilizing mutations in an ideal immunogen [46, 47]. Such stabilizing mutations present relevant Env conformations which bind bNAbs, but not non-bNAbs [46]. Specifically, Env with a single leucine to serine substitution in gp41 at position 669 (L669S) showed increased neutralization by MPER bNAbs 2F5 and 4E10, presumably via prolonged exposure of these relevant epitopes [48]. This longer period of epitope availability might be a desired attribute in an immunogen, a hypothesis that is being tested in clinical trials as a liposome-based vaccine (ClinicalTrials.gov Identifier: NCT03934541). We aimed to determine whether the Q563R-mediated effects on Env conformation affected neutralization by the rare bNAbs targeting the gp41 region [49–53]. MPER bNAbs 10E8, 7H6, 2F5, Z13e1 and 4E10 all demonstrated a 5- to 10-fold improvement in neutralization of the NE1 Q563R virus, compared to the NE1 virus (**Fig 6**). Similarly, bNAbs 10E8, 7H6 and 2F5 demonstrated 3- to 5-fold improvement in neutralization of the E1 virus, relative to the E1 R563Q virus (**Fig 6A, 6B and 6C**). The R563Q substitution in E1 R563Q rendered the virus >10-fold resistant to the MPER bNAb Z13e1 (**Fig 6D**). These data indicate that the Q563R change allows for better recognition by the rare, difficult-to-induce MPER bNAbs.

## Proposed model for the rescue of a Q563R-mediated infectivity defect by HR1-targeting antibodies

Based on these data, we propose a model in which the Q563R change in the Env E1 disrupts the HR1-HR2 interaction required for six-helix bundle formation (**Fig 7**). This results in a

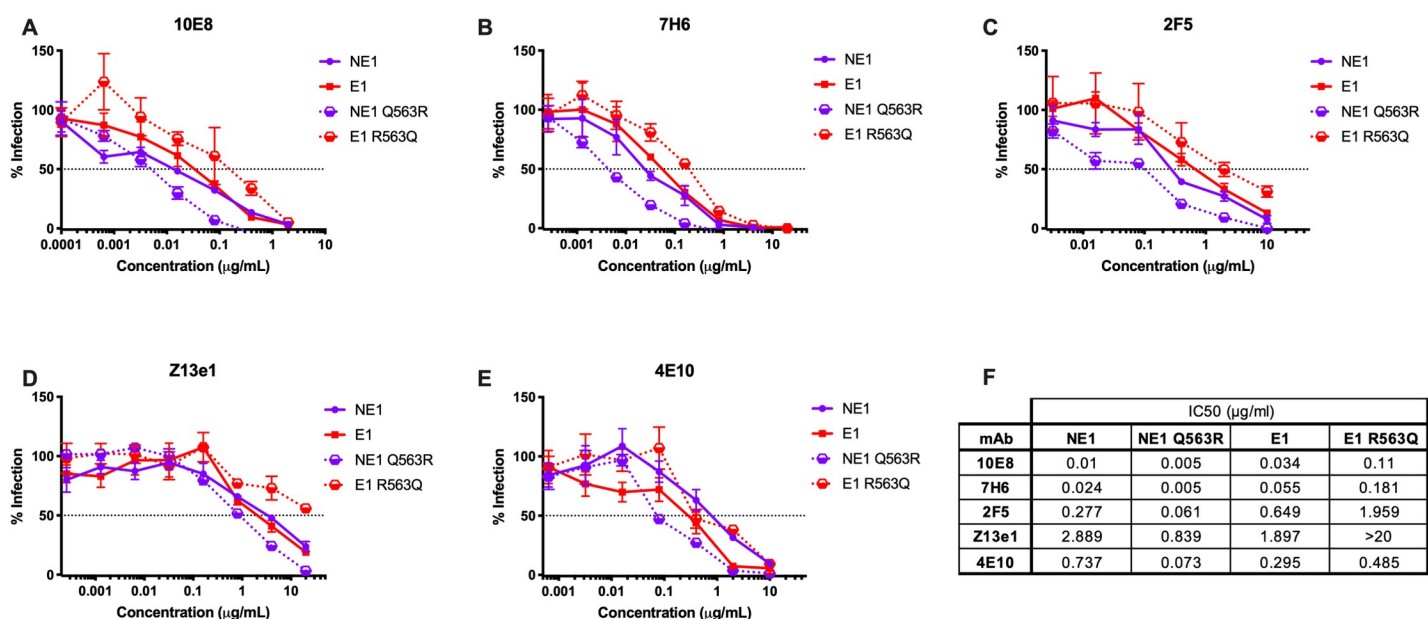

**Fig 6. Viruses with Q563R Envs exhibit increased sensitivity to neutralization by MPER bNAbs.** Infection of TZM-bl cells by NE1, NE1 Q563R, E1 and E1 R563Q viruses was tested in the presence of MPER-targeted bNAbs (**A**) 10E8, (**B**) 7H6, (**C**) 2F5, (**D**) Z13e1 and (**E**) 4E10, with (**F**) IC50 values as indicated. All assays were conducted in triplicate. Data are represented as mean values; error bars indicate SEM. The dotted line indicates 50% infection.

fusion-deficient gp41 conformation, leading to reduced infectivity. In turn, we hypothesize that gp41-targeting antibodies that bind the C-C loop serve to bring the HR1 and HR2 domains in proximity to one another or to stabilize the six-helix bundle. This stabilized six-helix bundle restores a fusogenic gp41 conformation and rescues Env infectivity.

## Discussion

HIV-1 entry, including binding of the CD4 receptor and coreceptors, is a tightly orchestrated process during which Env adopts numerous transient structures, leading to six-helix bundle formation and membrane fusion [14]. Proper interactions among critical Env residues are required for successful virus entry. Engineered mutational analysis has previously highlighted how changes in gp41 can reduce viral infectivity by disrupting interactions with gp120 or within gp41 [16, 17, 20, 54]. While gp41 changes do occur *in vivo*, not all of these are sufficiently viable, and many adaptive changes require compensatory mutations to enable successful infection. [24–26, 55, 56]. In contrast, this study identifies a mechanism by which HIV-1 utilizes a host immune response to overcome a naturally occurring viral defect in infectivity. We demonstrate that the reduced infectivity of viruses with the primary Env E1 clone is not due to decreased binding of CD4 or premature CD4-induced conformational changes, but rather from a defect in gp41-mediated fusion. We identify a rare arginine variant at gp41 residue 563 to be responsible for this reduced infectivity, and demonstrate the ability of gp41-targeted antibodies to alleviate this viral defect and restore infectivity. These data highlight the ability of HIV-1 to compensate for inherent viral defects by coopting host immune responses.

Viruses with a Q563R change in gp41 have been particularly difficult to study due to their low infectivity [22]. In our study, the NE1 Q563R virus was found to be 10-fold less infectious than the E1 virus containing the Q563R change, implying that in addition to the role of gp41 antibodies in rescuing the infectivity defect, the primary Env E1 likely contains other changes

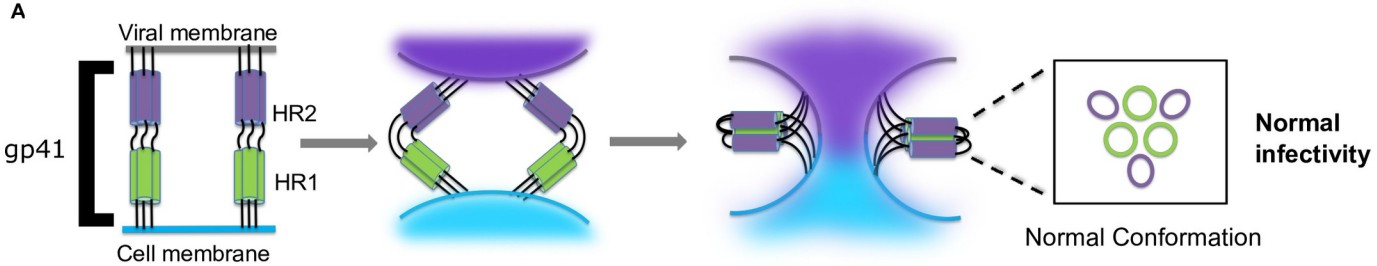

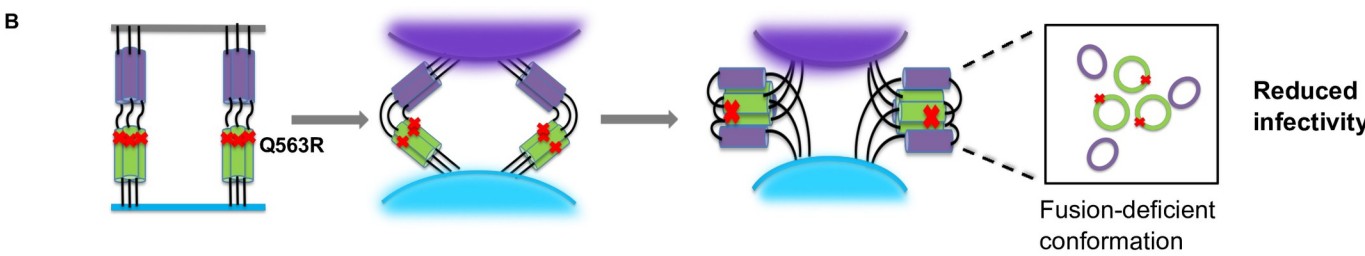

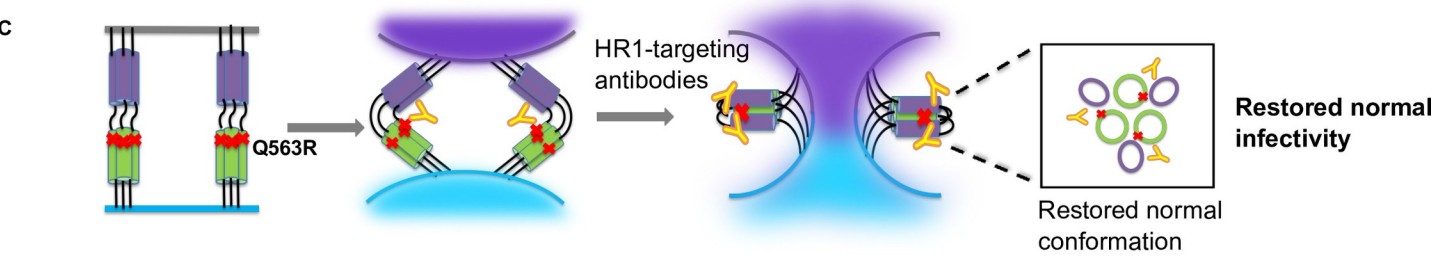

**Fig 7. Schematic representation of the proposed model. (A)** The top panel shows formation of the six-helix bundle and membrane fusion in normal HIV-1 infection. **(B)** Middle panel indicates the effect of the Q563R change in disrupting the six-helix bundle, resulting in an infectivity defect. **(C)** Bottom panel indicates restoration of infectivity by anti-HR1 targeting antibodies.

that have a compensatory effect. Indeed, Env E1 is divergent from Env NE1 at residues 605, 621 and 648 in gp41 (**S4 Fig**), which may serve to help restore some of the Q563R-mediated defects. As mentioned above, viral defects in HIV-1 have previously been reported to be compensated by other changes across the genome [56, 57]. Changes in gp41 have also been shown to modulate CD4 binding and CD4-induced conformational changes [16, 17]. However, we observed no change in CD4 neutralization or sensitivity to CD4i antibodies between the E1 and NE1 viruses. The role of residue 563 in receptor binding and associated conformational changes in different Env backbones is currently being tested.

Previous reports have identified mechanisms by which HIV-1, during the process of viral escape from selection pressures, gains dependence on these antiviral factors for infection and

replication *in vivo* [27, 58]. Zhou *et al.*, in studying the development of resistance to the capsid inhibitor PF74 identified a novel mutant virus with an impaired replication capacity in a patient treated with this inhibitor [58]. However, the replication of the mutant virus was surprisingly found to be stimulated by PF74. In another example, Baldwin *et al.* studied an HIV-1-infected individual that failed to respond to T20 treatment. They documented the emergence of a viral variant that not only developed resistance to the fusion inhibitor T20, but was highly dependent on T20 for viral infection [27]. These studies provide examples where through the development of resistance to synthetic antiviral compounds used to treat patients, viral fitness of these variants also becomes dependent on these antiviral compounds. Our study demonstrates, we believe for the first time, a mechanism whereby a viral defect is alleviated by the host's humoral immune response, specifically the Q563R-determined impaired infectivity of E1 viruses being restored in the presence of anti-HR1 antibodies. Our autologous neutralization assays rescuing the Q563R-determined viral defect highlight the approach of testing naturally occurring viral variants in the presence of autologous plasma, to accurately understand *in vivo* viral activity. These data thus demonstrate the importance of testing HIV-1 mutations and their effect on fitness in the context of an ongoing infection and the host immune response.

As previously mentioned, viral escape mutations are generally associated with a fitness cost. Env E1 was isolated from a late (6.5 ypi) plasma sample from individual 653116. Although R563 is a gp41 residue rarely identified in natural infection (www.hiv.lanl.gov), in individual 653116 R563 was first identified as a minority variant at 3.5 ypi. This suggests that this mutation in individual 653116 may have evolved in response to factors associated with ongoing infection, including escape from the host immune response. The association of escape from autologous antibody specificities and the emergence of R563 is currently being investigated.

Based on structural studies involving gp41-mediated fusion, HR1 residue 563 interacts with HR2 to stabilize the six-helix bundle. Glutamine at position 563 is hypothesized to interact with multiple residues within both HR1 and HR2. Q563 potentially forms hydrogen bonds via its polar side chain with L565 within the same HR1 and may influence the interaction of the adjacent residue Q562 with I559 [20]. Q563 also interacts with Q577 of the adjacent protomer within the six-helix bundle of trimers [16]. The association of Q563 with HR2 residues I642 and I646 is also implicated in the formation and stability of the six-helix bundle (**S7A Fig**) [20]. Q563 substitutions to alanine, methionine, glutamic acid or glycine have been studied for their effect on viral fitness. More common changes from glutamine to alanine or glycine that are smaller and more flexible minimally affect virus fusion, whereas changes to methionine or glutamic acid, which are larger amino acids, negatively affect viral infectivity to a greater degree [20]. A Q563R change represents a similar dramatic change in the properties of the amino acid at this position. This Q→R change introduces the long and bulky side chain containing the guanidinium group, which may destabilize the six-helix bundle by disrupting interactions of HR1 with HR2, as shown in **S7B, S7C and S7D Fig** [14]. The effect of a Q563R arginine substitution has been studied by engineering this mutation into NL4.3, HXB2 and YU2 laboratory strains, but efforts to study the viral entry of naturally occurring Q563R variants has been unsuccessful [16, 17, 59, 60]. This Q563R change has previously been shown to reduce infectivity to undetectable levels, making it impossible to study *in vitro* [22]. Naturally occurring Q563R variants are rare but have been found in infected individuals [61–65]. We hypothesize that the positively charged side chain of arginine disrupts six-helix bundle stability by repelling other positively charged residues within gp41. Additionally, the large guanidinium group at the end of the arginine side chain may be too bulky to allow for a stable six-helix bundle (**S7 Fig**).

The rescue of the Q563R variant by both relatively early (2 ypi) and late (6.5 ypi) autologous plasma, as well as by heterologous plasma from multiple individuals, corroborates findings that these gp41-specific antibodies develop early in HIV-1 infection [66, 67]. The gp41 ectodomain, which contains the HR1 and HR2 subunits, is highly conserved and immunodominant, accounting for the similarity in non-neutralizing gp41-specific antibodies from different individuals [68]. Notably, monoclonal antibodies 240D, 246D and F240 that alleviate the Q563R-mediated fusion defect bind a specific epitope within the disulfide loop in gp41. These antibodies mediate ADCC of infected cells via their Fc-receptors [69]. 240D, 246D and F240 have also been implicated in complement-mediated, antibody-dependent enhancement of infection [41]. Through the use of heat-inactivated plasma and FcR blockers, we have shown that the Q563R-specific increase in infection is not due to complement or FcR-mediated enhancement. Since the increased infection is seen specifically with Envs harboring a Q563R mutation, we conclude that these antibodies act to restore a viral defect rather than functioning by generally enhancing infection (**Figs 5 and S6**).

The Env trimer is a critical target for the development of vaccines to induce broadly neutralizing antibodies [45, 46]. A major obstacle to the induction of such broadly neutralizing responses is Env trimer instability, which conformationally masks critical antibody binding sites [70, 71]. For instance, the induction of MPER-specific bNAbs by vaccination has been especially difficult, largely due to the ambiguous structural information regarding native MPER conformation [72–74]. These findings have highlighted the importance of generating Env immunogens stabilized in different pre-fusion conformations, to allow for focused induction of neutralizing antibodies [75–78]. Some Env substitutions that stabilize the native gp160 trimer, with the goal of inducing neutralizing antibodies, have been enumerated by Torrents de la Peña A and Sanders RW [76]. Shen *et al.* demonstrated that the L669S change dramatically increased binding and neutralization by MPER bNAbs 2F5 and 4E10, suggesting that this change enhances exposure of 2F5 and 4E10 epitopes [48]. The incorporation of this alteration in a liposome-based vaccine currently in clinical trials exemplifies the strategy of conformationally stabilizing Env to present structurally relevant epitopes (ClinicalTrials.gov Identifier: NCT03934541). Along those lines, our study identifies a single-residue change in gp41 that appears to inhibit Env transitions into the six-helix bundle. The Q563R change may thus stabilize Env in a pre-hairpin intermediate conformation, which is the target of the potent and broad MPER antibodies 2F5, 10E8, 7H6 and 4E10 [79, 80]. Consistent with this model, we show that the Q563R change results in better virus neutralization by the rare and potent MPER bNAbs (**Fig 6**). Thus, the R563 substitution may stabilize Env in a prefusion conformation, to allow better binding of MPER antibodies. This study provides mechanistic insight into a novel gp41 change that may complement current approaches to immunogen design.

In summary, we demonstrate that a naturally occurring Q563R substitution in HIV-1 gp41 leads to a viral infectivity defect by inhibiting six-helix bundle formation and viral entry. We identify a novel phenomenon by which HIV-1 utilizes the host immune response to mitigate this viral defect. Furthermore, we propose a role for this Q563R substitution in stabilizing Env in conformations that might induce broadly neutralizing MPER-targeted antibodies more effectively.

## Materials and methods

### Ethics statement

All subjects gave written informed consent and the study was approved by the Massachusetts General Hospital Institutional Review Board.

## Study participants

Plasma samples were obtained from the Acute HIV Cohort and HIV Negative Cohort at the Massachusetts General Hospital in Boston, Massachusetts. Plasma samples from HIV-1 Clade B-infected individual 653116 were collected longitudinally up to fifteen years post infection. All samples used in this study were from time points in which the patient was not on antiretroviral therapy and was not diagnosed with AIDS. A total of 5 plasma samples from 3 patients (Ragon IDs 653116, 891439 and 269910) from the MGH Acute HIV Cohort were used in this study. Time points are defined from day of presentation with symptomatic acute HIV-1 infection. Plasma samples from individual 653116 are represented as "Autologous HIV+ Plasma" from either 2 years post infection (ypi), 4.6 ypi or 6.5 ypi. Plasma samples from individual 269910 (collected at 5 ypi) and individual 891439 (collected at 4.5 ypi), represented as "Heterologous HIV+ Plasma 1" and "Heterologous HIV+ Plasma 2", respectively. HIV Negative plasma was obtained from 2 individuals (Ragon IDs 129226 and 996837), represented as "HIV-Neg Plasma 1" and "HIV-Neg Plasma 2", respectively.

## Viral RNA isolation and quantification

Viral RNA was isolated from 1 mL of plasma as previously described [81]. Plasma was thawed at room temperature and centrifuged at 21,952 x g for 1.5 hours at 4˚C after which the pellet was re-suspended in 140 μl of the supernatant. The QIAamp Viral RNA Mini Kit (Qiagen) was used to isolate viral RNA per manufacturer's protocol with the exception of an additional on-column DNase treatment. Briefly, after the column was washed with Buffer AW1, 10 μl of DNase (Qiagen) diluted in 70 μl of RDD Buffer (Qiagen) was added to the column and was incubated for 15 minutes (min) at room temperature. The column was then washed again with Buffer AW1 prior to continuing with the manufacturer protocol. RNA was eluted in 50ul Elution buffer. Viral loads from the RNA isolation were determined using an HIV-1 Psi packaging signal qRT-PCR with primers F1 5'-CTC TCT CGA CGC AGG ACT CG-3' and R1 5'-GAC GCT CTC GCA CCC ATC TC-3'. The Quantifast SYBR Green RT-PCR (Qiagen) reagents were used per manufacturer's instructions in a 10ul total volume reaction and analyzed on a Lightcycler 480 (Roche).

## RT & PCR amplification

cDNA was synthesized using SSIII Reverse Transcriptase (Invitrogen). Purified vRNA was reverse transcribed to cDNA in a final volume of 20ul including 5 μl viral RNA, 1 μl of a deoxynucleoside triphosphate (dNTP) mixture (each at 2.5 mM), 0.5 μl antisense primer Env3Out (5'-TTGCTACTTGTGATTGCTCCATGT-3') at 10 μM, 4 μl 5X first-strand buffer, 1 μl dithiothreitol at 100 mM, 1 μl RNaseOUT (Invitrogen), and 1 μl SuperScript III reverse transcriptase. The reaction mixture was incubated at 50˚C for 60 min, followed by 55˚C for an additional 60 min. Finally, the reaction was heat inactivated at 70˚C for 15 min and then treated with 1 μl RNase H at 37˚C for 20 min. cDNA was used to amplify the envelope gene (*env*) by platinum-Taq Hi-Fi (Invitrogen) via nested PCR, using primers previously described [82]. The thermocycler conditions were 94˚C for 2 min, followed by 35 cycles of 94˚C for 15 s, 55˚C for 30 s, and 68˚C for 4 min, with a final extension of 68˚C for 10 min. The product of the first-round PCR was used as template in the second-round PCR under the same conditions for 45 cycles. Successful envelope amplifications were verified by agarose gel electrophoresis and products were purified using either QiaQuick PCR purification kit (Qiagen) or Pure Link gel extraction kit (Invitrogen), using the manufacturer's instructions.

## Single genome amplification and subcloning

Single genome amplification (SGA) was achieved using limiting dilutions in a nested PCR as described [82]. Briefly, cDNA was serially diluted and amplified to identify the dilution which would yield a PCR success rate of <30%; at such a dilution, most of the amplicons are generated from a single copy template. PCR was performed with 1x High Fidelity Platinum PCR buffer, 2mM MgSO4, 0.2 mM each dNTP, 0.2 uM each primer, and 0.1 units/uL High Fidelity Platinum Taq polymerase (Invitrogen) in a 20 uL reaction. Envelope amplicons were purified using either QiaQuick PCR purification kit (Qiagen) or Pure Link gel extraction kit (Invitrogen). These SGAs were reamplified by platinum-Taq Hi-Fi (Invitrogen) using gene specific primers. Amplification products were gel purified and cloned into the pEMC* expression vector using In-Fusion HD cloning (Takara). In-Fusion products were transformed into Stellar Competent Cells (ClonTech) and grown overnight on agar plates with ampicillin. Selected colonies were inoculated for DNA extraction by QIAPrep Spin Miniprep kit (Qiagen) and successful cloning was verified by Sanger sequencing (MGH CCIB DNA Core). Plasmid DNA from Sanger-verified preps was purified using the HiSpeed Plasmid Maxi Kit (Qiagen) or PowerPrep Plasmid Purification kit (Origene) and sequenced by Illumina.

## Illumina library construction and *env* sequencing

Samples were prepared for Illumina MiSeq using the Nextera XT Library Prep kit (Illumina). 1–1.25 ng of purified plasmid was tagmented and indexed as per the manufacturer's protocol. Indexed products were cleaned to remove small fragments via successive 70% and 60% SPRI using AMPure XP Beads (Beckman Coulter). The final product was quantified using a Promega Quanti-Flor fluorometer. Additionally, sample size was evaluated using the high sensitivity assay of the Agilent Bio-Analyzer. Tagmented amplifications were pooled and diluted to 12.5pM, then heated. The pooled library was spiked with 1% PhiX control (Illumina). The library was sequenced using the MiSeq v2 500-cycle kit (Illumina).

## Illumina data analysis

The adapter sequences were trimmed, and the library was demultiplexed on the MiSeq instrument (Illumina). Paired end reads were assembled into a contig using VICUNA *de novo* assembler and annotated with VFAT v1.0 [81, 83]. The envelope consensus sequences were then aligned to an HIV-1 Clade B reference using Mosaik 2.1.73 and variants were called using V-Phaser 2.0 [84]. Envelope sequences were submitted to GenBank (accession numbers MT023027- MT023033).

## Generation of *env* chimeras and point mutants

*Env* chimeras were generated by amplification of specific regions in *env* with Q5 High-Fidelity DNA Polymerase (New England Biolabs), using gene specific primers and subcloning using In-Fusion HD cloning (Takara). Successful cloning was verified, and DNA was prepped as described above.

## Cells, plasmids and antibodies

HEK293T/17 cells (ATCC) and TZM-bl cells were grown in Dulbecco modified Eagle medium (DMEM) with the addition of 10% fetal bovine serum (FBS), 1% penicillin-streptomycin, 1% L-Glutamine and 25 mM HEPES at 37°C and 5% CO2 content. TZM-bl cells were obtained through the NIH AIDS Reagent Program, Division of AIDS, NIAID, NIH: TZM-bl cells (Cat#8129) from Dr. John C. Kappes, and Dr. Xiaoyun Wu [85–89]. The HIV gp160

expression vector pEMC* was a generous gift from Dr. Nancy L. Haigwood (Oregon National Primate Research Centre). HIV-1 SG3 ΔEnv Non-infectious Molecular Clone was obtained through the NIH AIDS Reagent Program, Division of AIDS, NIAID, NIH from Drs. John C. Kappes and Xiaoyun Wu: [89, 90]. The following reagents were obtained through the NIH AIDS Reagent Program, Division of AIDS, NIAID, NIH: Human CD4-Ig Recombinant Protein from Dr. Xueling Wu [91], Human Soluble CD4 Recombinant Protein (sCD4) from Progenics, Anti-HIV-1 gp120 Monoclonal (17b) from Dr. James E. Robinson [1, 92–96], Anti-HIV-1 gp120 Monoclonal (2G12) from Polymun Scientific [52, 97–100], HIV-1 IIIB C34 Peptide from DAIDS, NIAID [101], D5 (HIV Env gp41) Monoclonal Antibody, Cat#12296 from Dr. Danilo Casimiro [38–40], Anti-HIV-1 gp41 Monoclonal (240-D) from Dr. Susan Zolla-Pazner [102, 103], Anti-HIV-1 gp41 Monoclonal (246-D) from Dr. Susan Zolla-Pazner [102–105], Anti-HIV-1 gp41 Monoclonal (F240) from Dr. Marshall Posner and Dr. Lisa Cavacini [42], T32 Monoclonal Antibody (Cat#11391) from Dr. Patricia Earl, NIAID [43], Anti-HIV-1 gp41 Monoclonal (98–6) from Dr. Susan Zolla-Pazner (cat# 1240) [102–105], Anti-HIV-1 gp41 Monoclonal (NC-1) from Dr. Shibo Jiang [106], Anti-HIV-1 gp41 Monoclonal (2F5) from Polymun Scientific (cat# 1475) [52, 107, 108], HIV-1 anti-gp41 mAb (10E8), from Dr. Mark Connors [49], Anti-HIV-1 gp41 Monoclonal (4E10) from Polymun Scientific [109].

## Additional reagents

The following reagents were obtained through the NIH AIDS Reagent Program, Division of AIDS, NIAID, NIH: RSC3 from Drs. Zhi-Yong Yang, Peter Kwong, Gary Nabel (cat #12042) [110, 111], HIV-1 MN gp41 Recombinant Protein from ImmunoDX, LLC. CD4mimetic DMJ-II-121 and anti-HIV-1 V2 Monoclonal antibody 902090 were generous gifts from Drs. Amos B. Smith III (University of Pennsylvania) and Barton Haynes (Duke University), respectively. CD4i antibody 447-52D was a generous gift from Dr. Michael Farzan (The Scripps Research Institute).

## Pseudovirus expression

Pseudotyped HIV-1 was produced by calcium phosphate transfection (Takara Bio Inc) of HEK293T cells (T75 flask) with 3.75 ug of pEMC* expressing the specific envelope glycoprotein and 11.25ug of an HIV-1 expression vector lacking a functional *env* gene (pSG3ΔEnv). Transfected cells were incubated at 37˚C / 5% CO2. The medium was changed 12 h post-transfection and collected 48 h later. Supernatants were passed through a 0.45-um syringe filter and stored at -80˚C.

## TZM-bl luciferase assay

Pseudovirus titers were determined as described [112]. Pseudovirus dilutions were incubated with TZM-bl cells for 48 hours at 37˚C. After 48 hours, cells were lysed with Britelite Plus (Perkin Elmer) and luciferase was measured using a TopCount Nxt plate reader (Perkin Elmer). Pseudovirus dilutions that yielded 100,000 relative luciferase units (RLU) readings were used for neutralization assays. TZM-bl neutralization assays were performed as previously described [113]. Briefly, HIV-1 pseudoviruses were preincubated with titrated amounts of antibody or heat-inactivated plasma (56˚C for 45 minutes) in DMEM with 10% FBS for 1 h at 37˚C. TZM-bl cells were detached by trypsinization, diluted in DMEM with 10% FBS and added to the pseudovirus-inhibitor mixture at 10,000 cells/well. Cells were incubated at 37˚C for 48 hours. After 48 hours, cells were lysed with Britelite Plus (Perkin Elmer) and luciferase was measured using a TopCount Nxt plate reader (Perkin Elmer). Percent infectivity (% infectivity) was calculated as RLU(Virus+EntryInhibitor)/RLU(VirusOnly) * 100. Fifty percent inhibitory

concentrations ($IC_{50}$s) were determined with GraphPad Prism 8.0 software using sigmoidal four-parameter logistic curve analysis.

## Gp41 ELISA

Enzyme-linked immunosorbent assay (ELISA) plates (Costar) were coated with 2 μg/mL HIV-1 MN gp41 recombinant protein and incubated overnight at 4˚C. Plates were washed twice with PBS plus 0.05% Tween 20 (PBS-T) and blocked with 1% bovine serum albumin (BSA) in PBS for 1 hour, at room temperature with shaking. Samples were washed 5 times with PBS-T before adding heat-inactivated plasma or fixed concentrations of VRC01 or F240 monoclonal antibodies. Plates were incubated for 3 hours, at room temperature with shaking. Samples were washed six times with PBS-T and labeled with a horseradish peroxidase-conjugated secondary antibody (Bethyl Laboratories Inc.) recognizing human IgG. Plates were incubated for 1 hour at room temperature with shaking and then washed 6 times with PBS-T. Tetramethylbenzidine (TMB) solution (Fisher) was added and left for 10 min at room temperature, with shaking. The reaction was stopped with 2M sulfuric acid ($H_2SO_4$). Plates were immediately read for absorbance at 450 nm with a Tecan plate reader.

## Fc receptor blocking

Human TruStain FcX (Fc Receptor Blocking Solution) (BioLegend) was used for Fc receptor blocking, as per the manufacturer's instructions. Briefly, TZM-bl neutralization assays using HIV-positive plasma were performed as described above. In this case, cells were incubated with 5 μl of Human TruStain FcX per million cells, mixed and incubated at room temperature for 5–10 minutes before adding to the pseudovirus-plasma mixture. Cells were incubated at 37˚C for 48 hours. After 48 hours, cells were lysed with Britelite Plus (Perkin Elmer) and luciferase was measured using a TopCount Nxt plate reader (Perkin Elmer). Fifty percent inhibitory concentrations ($IC_{50}$s) were determined with GraphPad Prism 8.0 software using sigmoidal four-parameter logistic curve analysis.

## Virus-cell fusion assay

The Blam-Vpr virus-cell fusion assay was performed essentially as described by the manufacturer (LiveBLAzer kit, ThermoFisher). HEK293T cells were transfected with plasmids encoding HIV-1 Gag-Pol, the Blam-Vpr fusion protein (AddGene 21950), and gp160 from the NE1, E1, NE1 Q563R, or E1 R563Q HIV-1 strains [36]. Virus was collected 24 hrs post-transfection and concentrated 10-fold by centrifugation over a 5% sucrose cushion. Pelleted virus was resuspended in phenol red-free DMEM medium supplemented with 10% FBS. The concentrated virus was passed to TZM-bl cells maintained in the same medium. Virus was allowed to bind the target cells by centrifugation for 30 mins at 3500 rpms at 4˚C. Unbound virus was removed by washing cells with cold HBSS and replaced with growth media. Virus entry was then allowed to proceed for 1.5 hrs at 37˚C. Following incubation, the medium was replaced with the CCF4-AM fluorophore in the loading solution provided in the Live-BLAzer kit and incubated overnight at 11˚C. Fluorescence was analyzed by excitation at 400 nm, and detected at 460 nm and 528 nm.

## Molecular modeling

Side-chain mutagenesis and subsequent rotamer analyses were done using the PyMOL Molecular Graphics System, Version 2.0 Schrödinger, LLC, with PDB 1AIK as a model [14].

## Supporting information

**S1 Fig. Depletion of HIV-1-positive plasma by RSC3core gp120 retains gp41-binding antibodies.** Binding of **(A)** gp41-specific antibody F240 and gp120-specific antibody VRC01 to plates coated with MNgp41 protein. MNgp41 binding of **(B)** autologous HIV-1-positive plasma (4.6 ypi) depleted of gp120 antibodies by RSC3core protein: "gp41 fraction", or gp120 antibodies eluted from the RSC3core protein "gp120 fraction" and HIV-1-negative plasma (control). Samples were tested in duplicate. Data represents mean values and error bars indicate SEM.
(TIF)

**S2 Fig. The V1/V2, C2, C3, V3, C4, V4 and C5 regions in Env E1 do not mediate HIV-1 plasma-dependent increased infectivity.** Pseudoviruses expressing **(A)** V1/V2 chimeras of Envs E1 and NE2 were tested for infection of TZM-bl cells in the presence of HIV-1-positive plasma (6.5 ypi). **(B)** Env chimeras of E1 and NE1 swapping the C2, C3, V3, C4, V4 and C5 regions were tested for infection of TZM-bl cells in the presence of HIV-1-positive plasma (4.6 ypi).
(TIF)

**S3 Fig. A Q563R single residue change supports increased Env infectivity with heterologous HIV-1 positive plasma.** Comparative infection of TZM-bl cells in the presence of a second heterologous HIV-1 positive plasma sample (Heterologous HIV+ Plasma 2) by NE1, NE1 Q563R, E1 and E1 R563Q viruses. All assays were done in triplicate. Data are represented as mean values; error bars indicate SEM. The dotted line indicates 50% infection.
(TIF)

**S4 Fig. gp41 alignment of NE1, NE2, NE3, NE4, NE5, NE6 and E1 Envs.** Amino acid alignment of NE1-NE6 and E1 is shown. HXB2 numbering is shown on the top left corner of each section. Dots indicate sequence identity. Non-conserved residues are displayed. The Q563R change unique to E1 is shown in red.
(TIF)

**S5 Fig. Anti-cluster I mAbs mediate increased infectivity of Q563R Envs.** Infection of TZM-bl cells by NE1, NE1 Q563R, E1 and E1 R563Q viruses was tested in the presence of anti-cluster II mAbs **(A)** 98–6, **(B)** NC-1 and V2-targeting antibody **(C)** 902090. **(D)** The gp41-binding epitopes of the HR1- and HR2-targeting antibodies tested for ability to increase infectivity.
(TIF)

**S6 Fig. Anti-HR1 mAb 246-D restores E1 infectivity to NE1 levels seen in the absence of antibody.** Fold change in infection of TZM-bl cells by E1 and E1 R563Q viruses with different amounts of 246-D is shown. All assays were done in triplicate. Data are represented as mean values; error bars indicate SEM.
(TIF)

**S7 Fig. Q563R potentially creates steric clashes within the six-helix bundle.** Potential interactions of **(A)** Q563 within HR1 (inner helix) with residues in HR2 (outer helix) are shown. Dotted lines indicate atomic distances between these residues. Potential steric clashes of Q563R with **(B)** isoleucine at position 642, **(C)** histidine at position 643 and **(D)** isoleucine at position 646 within HR2 are depicted. All images were created using the PyMOL Molecular Graphics System, Version 2.0 Schrödinger, LLC, using PDB 1AIK as template [14].
(TIF)

**S1 Data. Supporting numerical data.** Excel spreadsheet containing in separate sheets the underlying numerical data for Figs 1A, 1B, 1C, 1D, 1E, 1F, 1G, 1H, 1I, 2A, 2B, 2C, 2D, 2E, 2F, 3A, 3B, 3C, 3D, 4A, 4B, 4C, 4D, 5A, 5B, 5C, 5D, 5E, 5F,6A, 6B, 6C, 6D, 6E and S1A, S1B, S2A, S2B, S3, S5A, S5B, S5C, S6 Figs.
(XLSX)

## Acknowledgments

We would like to acknowledge the participants in the MGH Acute HIV Cohort and the HIV Negative Cohort as well as Dr. Eric Rosenberg, Dr. Marcus Altfeld, Sue Bazner and Graham McGrath for providing us with plasma samples. We would like to thank Dr. Nancy Haigwood for providing us with the pEMC* vector for Env expression. We thank Dr. Matthew Gardner and Dr. Michael Farzan for providing us with mAb 447-52D to test changes in CD4i conformation of Env. We also thank Dr. Daniel Claiborne for providing us with qRT-PCR primers and Dr. Christian Boutwell for insightful discussion on this manuscript.

## Author Contributions

**Conceptualization:** Vinita R. Joshi, Aaron G. Schmidt, Todd M. Allen.

**Data curation:** Ruchi M. Newman, Melissa L. Pack, Karen A. Power.

**Formal analysis:** Vinita R. Joshi, Todd M. Allen.

**Funding acquisition:** Todd M. Allen.

**Investigation:** Vinita R. Joshi, James B. Munro, Ken Okawa, Todd M. Allen.

**Methodology:** Vinita R. Joshi, Melissa L. Pack.

**Project administration:** Vinita R. Joshi, Karen A. Power, Todd M. Allen.

**Resources:** Karen A. Power, Navid Madani, Joseph G. Sodroski, Todd M. Allen.

**Software:** Ruchi M. Newman, Melissa L. Pack, Karen A. Power.

**Supervision:** Todd M. Allen.

**Validation:** Vinita R. Joshi.

**Visualization:** Vinita R. Joshi, Aaron G. Schmidt, Todd M. Allen.

**Writing – original draft:** Vinita R. Joshi, Todd M. Allen.

**Writing – review & editing:** Vinita R. Joshi, Ruchi M. Newman, James B. Munro, Navid Madani, Joseph G. Sodroski, Todd M. Allen.

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
