## [Decision Letter · Decision Letter 0]

11 Mar 2020

Dear Prof. Allen,

Thank you very much for submitting your manuscript "Gp41-targeted antibodies restore infectivity of a fusion-deficient HIV-1 envelope glycoprotein." for consideration at PLOS Pathogens. As with all papers reviewed by the journal, your manuscript was reviewed by members of the editorial board and by several independent reviewers. The reviewers appreciated the attention to an important topic. Based on the reviews, we are likely to accept this manuscript for publication, providing that you modify the manuscript according to the review recommendations.

Sincerely,

Guido Silvestri

Associate Editor

PLOS Pathogens

Susan Ross

Section Editor

PLOS Pathogens

Kasturi Haldar

Editor-in-Chief

PLOS Pathogens

orcid.org/0000-0001-5065-158X

Michael Malim

Editor-in-Chief

PLOS Pathogens

orcid.org/0000-0002-7699-2064

Reviewer Comments (if any, and for reference):

Reviewer's Responses to Questions

**Part I - Summary**

Reviewer #1: This study describes a novel HIV-1 envelope (Env) protein, cloned from an infected individual, that has the unusual property of mediating virus infection more efficiently in the presence of Env-specific antibodies present in this (and at least one other) infected individual. The Env protein mediates HIV-1 infection by binding to the virus receptor (CD4) and then to a coreceptor (generally CCR5). These binding events cause significant conformational rearrangements in Env that enable it to mediate fusion between the viral and cell membranes, allowing the viral genome to gain entry to the cytoplasm. As the molecule responsible for viral entry and a target for neutralizing antibodies, Env has been intensively studied over the years and much is known about its structure and function as well as the humoral immune response to Env induced by virus infection. Env is impressive in its ability to respond to selective pressures – it readily acquires resistance to antibodies and small molecules that target its functions. Sometimes, this highly adaptable protein evolves unusual traits. A small peptide that targets the gp41 region of Env and prevents its membrane fusion activity is in clinical use. About 15 years ago, an Env was described that was actually dependent upon this drug in order for it to mediate membrane fusion and virus entry: no drug, no membrane fusion. This study describes an Env that is somewhat analogous. A series of Envs were cloned from an individual who had been infected for a number of years. One of these exhibited a 300% increase in infection in the presence of plasma from this patient (and also in the presence of plasma from an unrelated, HIV infected patient). Curious. In the absence of plasma, this Env exhibited reduced function relative to other Env clones. As for the mechanism, this is an experienced group, and they know all of the tricks – the paper marches through in a logical fashion and shows that this unusual phenotype is due to the specific combination of a single amino acid change in the gp41 region of Env and the presence of antibodies commonly generated during infection (and that don’t really neutralize virus) that bind to gp41. The conclusion is that the mutation makes Env less fit, but that patient-derived antibodies alleviate this effect along with some compensatory mutations in gp41. This is a novel infection-enhancement mechanism. Antibodies to viruses can sometimes enhance infection by mediating binding to cell via Fc receptors (notably flaviviruses), but I am not aware of an instance in which an antibody, by binding to a viral entry protein, alleviates the impact of an otherwise deleterious mutation.

It is reasonable to ask whether this is just a single, relatively isolated example of a novel compensatory mechanism, or if this might be relatively common. In favor of this being a rare event is the fact the specific mutation studied here is quite rare and that other clones from this patient did not exhibit the plasma-enhancement phenotype. On the other hand, when viruses are isolated from patients or specific Envs are cloned, infections are typically not done in the presence of patient plasma, and so Envs that function poorly in the absence of host antibodies due to this or other antibodies might well be missed (i.e. discarded after being deemed poorly functional). Perhaps it might be wise to include autologous plasma more frequently in virus infection assays (suggested by the authors). The authors also note that the specific mutation identified here enhances viral sensitivity to neutralization by some well characterized, broadly neutralizing antibodies to gp41 and speculate that this mutation might be useful in vaccination strategies. Time will tell. I have a few minor comments:

I would be interested to know if infection enhancement is seen with a larger panel of plasmas - two were used in this study, but then in the discussion it is stated that plasma from multiple individuals had this effect. I suppose 2 is multiple by definition, but could the authors be more specific in the text?

When starting to sort out what antibodies might be responsible for the enhancement effect, the authors fractionated plasma by using magnetic beads coated with a form of gp120. The flow-through faction contained the enhancing activity and so the conclusion is that gp41 antibodies are responsible. While this eventually proved to be true, it is unlikely that any artificial Env construct would can all gp120 antibodies – it is probably more accurate to say that their results suggest that gp41 antibodies are the ones responsible (page 8).

Reviewer #2: This is a very intriguing study that indicates a novel aspect of immune evasion for HIV-1 that could be helpful for vaccine design.

The experimental design is compelling and the manuscript is very well written.

Reviewer #3: The manuscript entitled Gp41 targeted antibodies restore infectivity of a fusion -deficient HIV-1 envelope glycoprotein, by Joshi, Allen and colleagues describes a naturally occurring isolate of HIV that incorporates a rare mutation in the gp41 subunit of the envelope protein. The distinguishing feature of this viral isolate (termed E1) involves its dependence on anti env antibodies on viral replication. The authors show that in an in vitro TZM-bl infection assay, E1 fails to infect cells in an efficient way in the absence of autologous sera. However, the presence of sera rescues infection in a way that renders E1 substantially more infectious. The authors methodically map this “defect” to amino acid 563 in gp41 and a rare Q563R substitution. They then proceed to gather evidence that Q563R disrupts the six-helix bundle that mediates membrane fusion. The authors suggest that both polyclonal sera and gp41 mAbs are able to rescue infectivity by stabilizing the six-helix bundle in a way that compensates for the disruptive effect of arginine 563.

With this information in hand the authors then proceed to demonstrate a relative increase in sensitivity of arginine 563 envs to a panel of broadly neutralizing mAbs that target gp41. They argue that these increases indicate that the structural changes mediated by this rare substitution may provide insight into the design of env -based immunogens that might be used in an HIV vaccine.

**Part II – Major Issues: Key Experiments Required for Acceptance**

Reviewer #1: (No Response)

Reviewer #2: Very few issues were observed and should be barely considered as major.

1. The authors report that heterologous plasma could rescue the infectivity of the E1 R562Q isolates. Was the heterologous plasma characterized for the presence of Nab as suggested for the autologous?

2. Related to the above comment, we observed a statement in the discussion stating that " heterologous plasma from multiple donors" (line 424) was used. However, this is not reflected in any of the figures.

3. The independence from Fc Receptor should be supported by the utilization of Fab to enhance the infection.

Reviewer #3: Comments

1. The description of a mutation that relies on the host antibody responses to achieve efficient infection is quite interesting. The authors note that related phenomena have been described in the context of drug treatment (T-20). From the standpoint of the multitude of ways in which HIV can overcome host immune responses this observation stands out. It is also a bit discouraging in that it makes one wonder if there are almost unlimited ways in which the virus can evolve escape mechanisms. On line 381 the authors indicate that “for the first time” they have identified a mechanism whereby a viral defect is repaired by the host immune response. I would suggest that this statement be circumscribed to indicate more specifically that this a first in terms of a host humoral immune response that improves the function of the envelope.

2. All of the infection assays are carried out with TZM-bl cells, including those used to measure the sensitivity of the envs to gp41 mAbs. I am concerned that these observations might not reflect the impact of arginine 563 in a virus that is targeting primary CD4+ T cells. Given the substantial effort that went into identifying and characterizing this interesting mutation, I wonder why the authors don’t want to find out if their observations are relevant to an infection system that is a little more reflective or reality.

**Part III – Minor Issues: Editorial and Data Presentation Modifications**

Reviewer #1: (No Response)

Reviewer #2: (No Response)

Reviewer #3: (No Response)

PLOS authors have the option to publish the peer review history of their article (what does this mean?). If published, this will include your full peer review and any attached files.

Reviewer #1: No

Reviewer #2: Yes: Guido Ferrari

Reviewer #3: No
---

## [Decision Letter · Decision Letter 1]

24 Apr 2020

Dear Prof. Allen,

We are pleased to inform you that your manuscript 'Gp41-targeted antibodies restore infectivity of a fusion-deficient HIV-1 envelope glycoprotein.' has been provisionally accepted for publication in PLOS Pathogens.

Best regards,

Guido Silvestri

Associate Editor

PLOS Pathogens

Susan Ross

Section Editor

PLOS Pathogens

Kasturi Haldar

Editor-in-Chief

PLOS Pathogens

orcid.org/0000-0001-5065-158X

Michael Malim

Editor-in-Chief

PLOS Pathogens

orcid.org/0000-0002-7699-2064

Reviewer Comments (if any, and for reference):

Reviewer's Responses to Questions

**Part I - Summary**

Reviewer #1: Great response. I would not ask them to do additional experiments. First, they can't due to COVID. Second, it won't really change the already well documented central conclusion.

Reviewer #2: I can see the problems of making a Fab version of an Ab that binds the HR1 gp41 region.

The authors have provided data from an additional heterologous sample that support initial observations.

Considering constrains in laboratory activity - this is consider sufficient for acceptance of the manuscript.

Reviewer #3: This manuscript was returned to me after revision. One of the modifications that I requested was addressed. The second cannot be addressed because of the unplanned laboratory closing. My concern about not addressing the phenomenon described in the study in primary cells is something I hope the authors would consider in future studies, although I am not optimistic in this regard.

**Part II – Major Issues: Key Experiments Required for Acceptance**

Reviewer #1: (No Response)

Reviewer #2: None

Reviewer #3: na

**Part III – Minor Issues: Editorial and Data Presentation Modifications**

Reviewer #1: (No Response)

Reviewer #2: None

Reviewer #3: na

PLOS authors have the option to publish the peer review history of their article (what does this mean?). If published, this will include your full peer review and any attached files.

Reviewer #1: Yes: Robert Doms

Reviewer #2: Yes: Guido Ferrari

Reviewer #3: No

---

## [Editor Report · Acceptance letter]

1 May 2020

Dear Prof. Allen,

We are delighted to inform you that your manuscript, "Gp41-targeted antibodies restore infectivity of a fusion-deficient HIV-1 envelope glycoprotein.," has been formally accepted for publication in PLOS Pathogens.

Best regards,

Kasturi Haldar

Editor-in-Chief

PLOS Pathogens

orcid.org/0000-0001-5065-158X

Michael Malim

Editor-in-Chief

PLOS Pathogens

orcid.org/0000-0002-7699-2064